# Blockage of Store-Operated Ca^2+^ Influx by Synta66 is Mediated by Direct Inhibition of the Ca^2+^ Selective Orai1 Pore

**DOI:** 10.3390/cancers12102876

**Published:** 2020-10-06

**Authors:** Linda Waldherr, Adela Tiffner, Deepti Mishra, Matthias Sallinger, Romana Schober, Irene Frischauf, Tony Schmidt, Verena Handl, Peter Sagmeister, Manuel Köckinger, Isabella Derler, Muammer Üçal, Daniel Bonhenry, Silke Patz, Rainer Schindl

**Affiliations:** 1Gottfried Schatz Research Centre, Medical University of Graz, A-8010 Graz, Austria; linda.waldherr@medunigraz.at (L.W.); romana.schober@jku.at (R.S.); tony.schmidt@medunigraz.at (T.S.); 2Institute of Biophysics, JKU Life Science Centre, Johannes Kepler University Linz, A-4020 Linz, Austria; adela.tiffner@jku.at (A.T.); matthias.sallinger@jku.at (M.S.); irene.frischauf@jku.at (I.F.); isabella.derler@jku.at (I.D.); 3Centre for Nanobiology and Structural Biology, Academy of Sciences of the Czech Republic, 373 33 Nové Hrady, Czech Republic; mishrad@nh.cas.cz; 4Department of Neurosurgery, Medical University of Graz, A-8010 Graz, Austria; verena.handl@medunigraz.at (V.H.); muammer.uecal@medunigraz.at (M.Ü.); 5Institute of Chemistry, University of Graz, Heinrichstraße 28, A-8010 Graz, Austria; peter.sagmeister@uni-graz.at (P.S.); manuel.koeckinger@uni-graz.at (M.K.)

**Keywords:** STIM, Orai, Ca^2+^, SOCE, Synta66, pore, binding, pocket, docking, glioblastoma multiforme, GBM

## Abstract

**Simple Summary:**

Store-operated calcium channels constituted from the proteins Orai and STIM are important targets for development of new drugs, especially for the treatment of auto-immune diseases. Also, interference with channel function is linked to reduced cancer cell progression, making these channels potential targets for anti-cancer drug development. Therefore, inhibitors need to be evaluated for both their binding selectivity and their potential to interfere with cancer progression. Here, we investigated the inhibitor Synta66 and determined its site of binding via both patch clamp recordings and computational approaches and evaluated its potency as anti-cancer agent in glioblastoma multiforme cells. Our findings show that Synta66 is a highly selective ligand to the Orai1 pore and efficiently blocks store operated calcium entry in glioblastoma cells. Still, in the tested cell lines, Synta66 did not reduce cell viability. We therefore suggest Synta66 as a precise tool to observe interference of store-operated Orai1 channel function *in vitro* and of resulting downstream effects.

**Abstract:**

The Ca^2+^ sensor STIM1 and the Ca^2+^ channel Orai1 that form the store-operated Ca^2+^ (SOC) channel complex are key targets for drug development. Selective SOC inhibitors are currently undergoing clinical evaluation for the treatment of auto-immune and inflammatory responses and are also deemed promising anti-neoplastic agents since SOC channels are linked with enhanced cancer cell progression. Here, we describe an investigation of the site of binding of the selective inhibitor Synta66 to the SOC channel Orai1 using docking and molecular dynamics simulations, and live cell recordings. Synta66 binding was localized to the extracellular site close to the transmembrane (TM)1 and TM3 helices and the extracellular loop segments, which, importantly, are adjacent to the Orai1-selectivity filter. Synta66-sensitivity of the Orai1 pore was, in fact, diminished by both Orai1 mutations affecting Ca^2+^ selectivity and permeation of Na^+^ in the absence of Ca^2+^. Synta66 also efficiently blocked SOC in three glioblastoma cell lines but failed to interfere with cell viability, division and migration. These experiments provide new structural and functional insights into selective drug inhibition of the Orai1 Ca^2+^ channel by a high-affinity pore blocker.

## 1. Introduction

The Ca^2+^ channel protein complex formed by stromal interaction molecule (STIM) and Orai has been shown to mediate store-operated Ca^2+^ (SOC) influx in a large variety of cells including numerous cancer cell types (as reviewed in [1] and recently reported in [2,3]. Under physiological conditions, this ubiquitous Ca^2+^ signalling pathway is activated by stimuli that deplete Ca^2+^ from endoplasmic reticulum (ER) stores [4,5,6,7], which is sensed by STIM1 and STIM2 via a luminally located EF-hand [8,9,10,11,12]. In resting cells, ER Ca^2+^ concentrations are in the sub-millimolar range (~ 0.8 mM). When ER [Ca^2+^] is lowered to 100-400 µM [7,9,10], STIM1 becomes sequestered into puncta at the ER-plasma membrane (PM) junctions [7,13]. STIM2, on the other hand, can be activated by even smaller changes in ER [Ca^2+^] [9,14]. As well as clustering into puncta, STIM1 and STIM2 bind directly to the Orai channel family members, Orai1, 2 and 3 in the PM [15,16,17,18,19]. STIM/Orai coupling, in turn, activates the channel and mediates Ca^2+^ influx across the PM [20,21,22]. Physiologically, STIM1 and Orai1 are essential for immune responses such as T cell activation and mast cell degranulation [23]. In the brain, STIM1 and Orai1 regulate store-operated Ca^2+^ entry (SOCE) in astrocytes, which directly triggers slow vesicular release of glial transmitters, including ATP [24]. Electrophysiological recordings in glioblastoma cells [25] and in astrocytes [24] demonstrated that STIM1 and Orai1-mediated SOCE results in highly inward rectifying Ca^2+^ currents.

Interference with STIM/Orai channel signalling (and many other ion channels) has been shown to inhibit cancer cell progression (reviewed in [26,27]). SOCE inhibition might therefore represent an alternative treatment approach for cancers that are hard to treat with conventional methods, such as glioblastoma multiforme (GBM) [28,29,30]. The efficacy of pharmacological treatment of GBM has remained largely unchanged for several decades, mainly due to the limiting character of the blood–brain barrier which renders the use of many anti-cancer drugs impractical for GBM therapy [28,29,30]. 

siRNA-mediated silencing of STIM1/Orai1 expression or pharmacological SOCE inhibition in GBM cells have been shown to suppress cell growth [31,32], migration [33] and invasion [25], supporting the investigation of Ca^2+^ signalling blockers as potential novel therapeutic tools for GBM treatment. Based on this quest, several targeted pharmacological inhibitors of cellular Ca^2+^ signalling (for instance the imidazole derivative SKF-96365, diethylstilbestrol (DES) and 2-aminoethoxydiphenyl borate (2APB)), have been shown to efficiently inhibit store-operated Ca^2+^ entry pathways, suppress cell migration and increase apoptosis in various GBM cell lines [25,31,32,34]. The variable impact of these SOCE inhibitors on cell proliferation and viability amongst the GBM cell lines used [31,32,34], raises the question whether the elicited responses were the direct result of specific store-operated Ca^2+^ channel inhibition [35] or of cumulative effects. SKF-96365, for instance, also inhibits a variety of other ion channels, including K^+^ channels, whilst having stimulatory effects on non-selective ion channels [35,36]. 2-APB, similarly, inhibits Orai1, but potentiates Orai3-mediated currents [37,38,39]. 

The compounds Synta66, GSK-5503A, GSK-7975A and RO2959 are the highest affinity SOC channel inhibitors currently available [40,41,42,43,44]. Notably, novel SOC inhibitors developed by CalciMedica and Synta have completed Phase I human clinical trials for the treatment of moderate-to-severe plaque psoriasis (CM2489). CM4620, a compound with structural similarities to Synta66, is undergoing Phase II clinical trials to treat pneumonia in COVID-19 patients and acute pancreatitis [41,42,44]. Another Synta compound, PLRCL-2, has entered a Phase II clinical trial in plaque psoriasis. Whether Synta66 served as lead substance for the screening of PLRCL-2 and its structure has not yet been reported.

The present study provides a structural model of the inhibitory function of Synta66 and its direct binding close to the Orai1-selectivity filter. Mutations that affect the selectivity of Orai1 but also permeation of Na^+^ instead of Ca^2+^, diminished Synta66-mediated ability for inhibition of the Orai1 channel. Our experiments showed efficient blockage of SOC signaling in A172, LN-18 and U-87 MG GBM cell lines by Synta66. This blockade, did not, however, affect cell viability and even increased cell migration slightly, in contrast with previous studies using less-specific inhibitors (2APB, DES and SKF-96365) or gene silencing to suppress SOCE in GBM cell lines.

## 2. Results

### 2.1. Synta66 is a Direct Blocker of the Orai1 Pore

We first evaluated whether Synta66 inhibits the Orai1 channel directly by performing patch clamp experiments. To obtain directly detectable SOC current levels, we co-expressed STIM1 and Orai1 in HEK293 cells (Figure 1A). STIM1/Orai1-overexpressing HEK cells exhibited highly inward rectifying Orai1 currents upon passive ER Ca^2+^ depletion in patch clamp experiments (20 mM ethylene glycol-bis(β-aminoethyl ether)-*N*,*N*,*N*′,*N*′-tetraacetic acid (EGTA) in the pipette solution, Figure 1A,B, black trace). Addition of 10 µM Synta66 completely blocked STIM1/Orai1 currents at the maximum activation point, (Figure 1A, black trace) in line with our previous results [40]. Patch clamp experiments with 1 µM Synta66 treatment upon maximum STIM1/Orai1 mediated currents resulted in slow inhibition (Appendix A). Previous results have shown that over-expressed STIM1/Orai1 currents were inhibited with an inhibitory concentration (IC_50_) of ∼4 μM and a Hill coefficient of ∼1 [40]. A similar inhibitory profile has been reported for Synta66 on RBL mast cells exhibiting an IC_50_ of ∼1–3 μM and a Hill coefficient of 1.1 [45,46]. To determine whether Orai1 is directly targeted by Synta66, we disrupted the ring of glutamates that form the selectivity filter of the protein [21,47,48]. Conversion of a single glutamic acid to an aspartic acid (E106D) in each Orai subunit caused a loss of Ca^2+^ selectivity. Store-operated STIM1 and Orai1-E106D-mediated currents in transfected HEK cells were only partially inhibited to Synta66 treatment (10 µM, Figure 1A, red trace, and 1 µM Appendix A) and completely blocked by subsequent La^3+^ treatment (10 µM, Figure 1A, red). In a comparison to the reversal potentials of wild-type Orai1 (Figure 1B) and the Orai1-E106D mutant (Figure 1C, red), the latter exhibited a strong left-shift to ~0 mV, as previously described [21,47,48]. Treatment of the Orai1-E106D mutant with Synta66 (10 µM) did not significantly affect reversal potential (~0 mV, Figure 1C, blue trace). Computational docking experiments were performed to study the observed role of the Orai1 selectivity filter (Figure 1D–F). In order to determine favourable interaction sites for Synta66 in the Orai channel, we calculated glide scores for static Synta66 molecules at different positions in the pore entrance (Figure 1F and Appendix A). The most negative glide score indicates the highest protein-ligand affinity. The results obtained showed that Synta66 has high affinity for the loop1 and loop3 regions close to the Orai pore (Figure 1D, top view; Figure 1E, side view). Binding of Synta66 to the Orai pore selectivity filter is, thus, essential for the inhibition of SOCE currents, which is also in line with previous results with GSK-7975A and GSK-5503A [40]. The pose one (Figure 1F) with the most negative glide score (−5.162) was further investigated by Molecular Dynamic (MD) simulations (Figure 2).

### 2.2. Synta66 Binds Close to the Selectivity Filter and Extracellular Loop Region

Initial MD simulations that positioned three Synta66 molecules close to the pore and not in contact with the protein (Figure 2A) showed them to reach the pocket within 300 ns (Figure 2B,C). As each Synta66 molecule interacted with two Orai1 subunits, we reasoned that the hexameric channel could accommodate a maximum of three molecules (Figure 2A). This ligand density, however, appeared to hinder proper docking, as the molecules remained around 8-10 Å away from the selectivity filter (Figure 2C). 

Three Synta66 molecules positioned to pose 1 at the start of MD simulations remained within the binding pocket (Appendix A). It thus appears more likely that the channel accommodates fewer Synta66 molecules. Indeed, in simulations with a single Synta66 molecule at the starting pose 1 (Figure 2D, glassy cyan blue stick model), the compound was seen to anchor itself deeper within the transmembrane domain of the channel and to localize in much greater proximity to the selectivity filter after 100 ns (Figure 2D,E; Synta66 shown as sphere model at end position). Specifically, Synta66 interacted most frequently with the extracellular transmembrane helices TM1 and TM3 and connecting loops 1 and 3 (Figure 2F).

It should be noted that, whilst Synta66 diffused to the loop1/loop3 region in MD simulations, its end position was not identical to that determined by docking experiments (Appendix A). These experiments, nevertheless, consistently showed that both the extracellular TM1/TM3 helix region, close to the selectivity filter, as well as loops 1 and 3 of Orai1, contribute to Synta66 binding.

### 2.3. Inhibition of Orai1 Currents by Synta66 is Impaired by Ion Hydration

To evaluate the role of Orai1 loops 1 and 3 in Synta66 binding (Figure 3A), we generated an Orai1 loop1 mutant (L109D, H113G, Y115G, Figure 3A–C) and an Orai1 loop3 mutant (F199G, P201G, L202G, Figure 3A,D,E) in order to directly interfere with Synta66 inhibition. Both mutants exhibited store-operated activation and non-selective currents when co-expressed with STIM1 (Figure 3C,E). Addition of Synta66 resulted in partial inhibition (Figure 3B,D). We reasoned that the shift from Ca^2+^ selective currents to non-selective currents may have been a direct impact of Synta66 inhibition. We previously found that an Orai1-H134A (Figure 3F) mutation resulted in constitutively active currents in the absence of STIM1 and store-depletion [22]. Co-expression of STIM1 and Orai-H134A shifted the reversal potential only by ~ −8 mV leftwards, demonstrating that this mutant retained a relatively high Ca^2+^ selectivity (Figure 3I), as previously described [22]. Synta66 (1 and 10 µM) partially inhibited Ca^2+^ currents in the plateau phase (Figure 3H,I and Appendix A). 

Permeation in these three Orai1 mutants is mediated by Na^+^ as well as by Ca^2+^ ions. Since non-selective, Na^+^-based currents do not bind a water shell as efficiently as Ca^2+^ ions, we directly evaluated whether the hydration of ions interacting with the Orai1-selectivity filter has an impact on Synta66 channel inhibition. After maximum activation of STIM1 and wild-type Orai1-mediated currents, we replaced the extracellular solution with a Na^+^-based divalent ion-free solution (Figure 3J–L). In line with previous results, the Orai1 pore was less selective to Na^+^ mediated currents than Ca^2+^ currents (Figure 3K). Remarkably, Na^+^ currents were, indeed, only partially inhibited by Synta66 (Figure 3J,K). Analogous time-course experiments with Synta66 in a standard Ca^2+^ solution, in contrast, resulted in complete inhibition (Figure 3L) as shown in Figure 1A.

These findings clearly demonstrate that Orai1 mutants with diminished Ca^2+^ selectivity are only partially inhibited by Synta66. Our results, therefore, suggest that ion hydration at the selectivity filter plays a critical role in Synta66-mediated Orai1 inhibition. 

### 2.4. Effect of Synta66 on Orai1 Pore Hydration

We and others have previously shown that hydration of the narrow Orai1 pore plays a crucial role in ion permeation [22,49,50]. In MD simulations, Synta66 molecules were docked to Orai1 wild-type and mutant channels at pose 1 (Appendix A). Pore hydration was then explored by estimating the number of water molecules within each pore during the last 50 ns of 200 ns simulations by calculating the axial position of the water molecule oxygen atom within a 10 Å radius from the pore centre. The lowest level of hydration was found for the Orai1 wild-type pore (Appendix A). Interestingly, a slight increase in hydration was noted for the Orai1-E106D mutant (Appendix A), while Orai1-H134A showed the highest hydration level (Appendix A).

Whilst an increase in the level of hydration concomitant with the constitutive activity of Orai1-H134A was observed, at the time scale of these simulations all variations of the Orai1 channels appeared to be unaffected by the presence of Synta66.

### 2.5. Synta66 Does Not Affect STIM1 Di-/Oligo-Merization 

We evaluated whether Synta66 has additional target sites in the store-operated Ca^2+^ entry pathway. In our previous patch-clamp experiments, STIM1 was co-expressed with Orai1 channel and mutants. Hence, we evaluated if Synta66 has an impact on the STIM1 activation cascade by fluorescence resonance energy transfer (FRET) confocal microscopy. Co-expressed cyan and yellow fluorescent protein (CFP/YFP)-tagged STIM1 yielded a tubular localization in resting HEK293 cells (Figure 4A). Calculated FRET interaction was clearly visible for STIM1 already at resting conditions, in line with previous results [16]. ER Ca^2+^ store-depletion, using Thapsigargin (TG) (1 µM) treatment, clearly increased STIM1 interactions and reached a maximum FRET value within 2 minutes (Figure 4B). The increased STIM1 interaction correlated well with a translocation of STIM1 into puncta (Figure 4A). In analogous experiments, with cells pre-treated for 20 min with Synta66 (10 µM), a similar STIM1 puncta translocation and FRET time-course was observed (Figure 4B,C). These experiments determine that Synta66 neither acts significantly on STIM1 localization nor on puncta formation.

### 2.6. STIM1-Orai1 Coupling is Unaffected by Synta66

As a second activation step, we also investigated the interaction of STIM1 with the Orai1 channel. We used C-terminally tagged CFP-STIM1 with C-terminally tagged YFP-Orai1 to monitor for a cytosolic interaction in confocal FRET microscopy. STIM1 and Orai1 localization was clearly distinct in resting cells (Figure 5A), concomitant with low FRET values (Figure 5B). TG treatment (1 µM) resulted in co-localization and puncta formation of STIM1 and Orai1 (Figure 5A) and FRET maximum values within 2 min (Figure 5B). Analogous experiments, with cells pre-treated with Synta66 for 20 min yielded a FRET time-course (Figure 5B) and translocation into puncta (Figure 5C) similar as in control experiments. These results determine that STIM1 activation mechanism and interaction with Orai1 remains unaffected by Synta66.

### 2.7. Calcium Recordings in Synta66-Treated GBM Cell Lines

To investigate the effects of Synta66-mediated SOCE inhibition in GBM cells, we examined three commonly used GBM cell lines (U-87 MG, A172 and LN-18). Orai1 and STIM1 expression in these cell lines was determined by Western blot analysis of deglycosylated cell lysates (for Orai1) and standard lysates (for STIM1). The highest level of Orai1 expression was observed in LN-18 cells, followed by A172 and U-87 MG (Figure 6A). STIM1 expression was also confirmed in all three lines (Figure 6B, Appendix A).

Endogenous SOC is intrinsically difficult to measure electrophysiologically in GBM cells owing to the reported small magnitude of Ca^2+^ currents in these cells (< 0.5 pA/pF), which prevent direct detection [25]. Ratiometric Ca^2+^ imaging with Fura-2 AM loaded cells was consequently used to determine relative SOCE levels (Figure 6C–F). GBM cells were initially monitored in a nominally Ca^2+^-free solution prior to addition of benzohydroquinone (BHQ, 30 µM) to inhibit ER-localized sarcoplasmic/endoplasmic reticulum calcium ATPase (SERCA) pumps and, thereby, generate transiently higher cytosolic Ca^2+^ concentrations. Ca^2+^ was afterwards added to the medium to determine SOCE activation (Figure 6C–F, black). ER Ca^2+^ depletion was also observed in the presence of Synta66 (1 µM in Appendix A, 10 µM in Figure 6), but SOCE was lacking in all three GBM cell lines (Figure 6C–F, blue), as determined by time-course experiments and by statistical analysis of relative SOCE levels (Figure 6C). 

### 2.8. Synta66 Moderately Enhances Growth of LN-18 and U-87 MG GBM Cells

Pharmacological blockage of SOCE has been previously shown to significantly impair the proliferation of a variety of GBM cell lines (C6, D54, U373, U-87 MG and U-251 MG) [31,32]. Here we evaluated the effect of Synta66 on the proliferation of A172, LN-18 and U-87 MG cell lines at 24, 48 and 72 h after treatment by quantification of Hoechst-stained nuclei (Figure 5). Proliferation of all three GBM cell lines treated with 1 µM Synta66 was comparable to that of vehicle-treated (0.1% DMSO) controls (Figure 7, light blue). Treatment with 10 µM Synta66, however, surprisingly enhanced proliferation of LN-18 and U-87 MG cells, measured at 48 h and 72 h (Figure 7B,C, dark blue), although the differences were only statistically significant for LN-18 at 48 h ( *: *p* < 0.05, Figure 7B). A172 cell division was not noticeably affected by Synta66 treatment (Figure 7A).

### 2.9. Synta66 Synergizes Moderately with TMZ Treatment in U-87 MG Cells

We next determined the effect of Synta66 treatment for 24, 48 and 72 h on the viability of A172, LN-18 and U-87 MG cells. The viability of all three lines was not detectably altered by treatment with either 1 µM or 10 µM Synta66 at any of the time points (Appendix A). When applied in combination with temozolomide (TMZ), a standard chemotherapeutic drug used routinely in clinical GBM handling, however, 10 µM Synta66 treatment increased TMZ-sensitivity of U-87 MG cells to a statistically significant extent (Appendix A). A Synta66-mediated increase in TMZ sensitivity was not significantly different in A172 cells (Appendix A). LN-18 cells, on the other hand, did not exhibit any change in TMZ-sensitivity upon Synta66 treatment (Appendix A). 

### 2.10. Synta66 Effects upon Cell Migration

Pharmacological interference with SOCE in various GBM cell lines has also been shown to significantly impair cell migration [32,33]. We evaluated whether Synta66 can influence the migratory potential of the GBM cell lines used in this study using a so-called scratch assay. This simple method monitors the rate of recolonization of a linear gap scratched in a confluent cell monolayer cultured in low serum culture media (1% foetal bovine serum (FBS)). Treatment of LN-18 cells with 10 µM Synta66, however, resulted in a rate of gap closure similar to that for vehicle-treated controls (Appendix A).

## 3. Discussion

Clinical trials of SOCE inhibitors have, to date, addressed the treatment of psoriasis, acute pancreatitis and pneumonia [44,52]. Pre-clinical data indicate that targeting the SOCE signalling cascade may also interfere with tumour growth and progression. Genetic silencing of STIM1/Orai1 and STIM1/Orai3 protein complexes by siRNA interference has, for example, yielded promising reductions in tumour progression and metastasis in xenograft models of breast, cervical and prostate cancer [53,54]. Research, stimulated by these and other findings, targeting SOCE in cultured GBM cells, has documented inhibition of cell growth, migration and proliferation by the SOCE inhibitors DES, SKF-96365 and 2APB [25,31,32,34]. The underlying mode of action of the inhibitors was, though, unclear: 2APB, for example, inhibits Orai1-mediated currents but instead stimulated Orai3 currents [37], SKF-96365 is a weak blocker (IC_50_ in µM range) and less selective [55], and DES is an important non-steroidal oestrogen modulator [56]. Amlodipine-mediated SOCE activation has, moreover, recently been shown to markedly reduce the viability of LN229 glioblastoma cells [57]. SOCE activation was stated to be a secondary result alongside the well-documented inhibition of voltage-gated Ca^2+^ channels by amlodipine [57]. 2-APB can additionally reverse STIM1 clusters [58,59,60]. Generally speaking, all four compounds have multiple, concentration-dependent target sites [35]. 

The present study investigated whether the above-described effects on GBM cells were direct results of SOCE inhibition, by a) studying the molecular mechanisms and specificity of Synta66 binding to the SOC complex and, b) observing the direct effects of Synta66-mediated SOC inhibition in three GBM cell lines. Using STIM1/Orai1-overexpressing HEK cells treated with 10 µM Synta66, we achieved an efficient block of Orai1 currents. SOCE inhibition by Synta66 was largely abolished by targeted mutation of the glutamate ring structure of Orai1 (E106D) that forms the protein’s selectivity filter. An evaluation of Orai1 mutants comprising docking experiments, MD simulations and patch-clamp results localized the Synta66 binding site to the extracellular TM1/3 helices and extracellular loops 1 and 3. Mutations affecting the loop1/loop3 pore entrance diminished Orai1 inhibition by Synta66. 

It is of note that inhibition of Orai channels by several SOC inhibitors, including Synta66, BTP2, GSK-7975A is dependent on the Orai isoform [61]. A recent work determined that 10 µM Synta66 inhibited Orai1 currents, while Orai2 currents were even further stimulated and Orai3 currents were largely unaffected [61]. We compared the amino-acid sequence within the three Orai isoform that were observed to dock with Synta66. L109, H113 and Y115 residues in Orai1 loop1, are conserved in the Orai3 channel, while H113 is a tyrosine in Orai2. This Orai1 loop1 binding site overlaps also with the Ca^2+^ accumulation region at the pore entrance [62]. F199 and P201 in TM3 helix of Orai1 are fully conserved in Orai2 and Orai3. L202 in Orai1 remains conserved as a hydrophobic amino acid, with a valine in Orai2 and an isoleucine in Orai3. These sequence analysis shows that key residues are largely conserved, however, additional interaction sites with Synta66 with Orai2 or Orai3 residues may further tune channel modulation.

Concatenated Orai hetero-dimers largely diminished Synta66 blocking [61], suggesting that the here investigated SOC channels in GBM cell lines are mainly based on Orai1 channels. In prostate cancer cells, it is of note that Orai3 expression is the only Orai isoform that is upregulated in comparison to healthy tissues [63]. A molecular switch from Orai1 homomers into Orai1-Orai3 heteromers resulted in store-independent currents and promoted cell proliferation [63]. Store-operated Orai1-Orai3 heteromers yield only a mild inhibition by Synta66 [61]. Targeting store-independent pathways by Orai1-Orai3 heteromers is not been addressed in this work, and provides an additional strategy for GBM treatment [63].

Hydrogen bond formation plays a key role in the docking of Orai1 channel pore inhibitors (present study and [64]). In line with this, a docking analysis of 55 SOCE inhibitors determined major interactions between these compounds and Orai1 residues Q108, H113 and D114 [64]. The range of glide scores for the panel of tested compounds is, furthermore, comparable to our approach. Orai1 mutations that only shifted Ca^2+^ to Na^+^ permeation, notably affected the efficiency of inhibition by Synta66. Even permeation of Na^+^ through the wild-type Orai1 pore instead of Ca^2+^ ions diminished Synta66-mediated inhibition. This dominant inhibitory effect of Synta66 on the selectivity filter region, does not, however, rule-out additional binding sites within Orai channels [44]. Instead, neither STIM1 puncta formation nor STIM1-Orai1 coupling was significantly affected by Synta66 (in this study) nor GSK-7975A [40]. 

To bridge the results obtained with Orai1-overexpressing transfected HEK293 cells to GBM cells, we monitored SOCE by Ca^2+^ imaging. STIM1 and Orai1 have been previously reported to play a major role in SOCE in GBM [25]. Low-level SOCE in GBM cells [25], however, greatly complicates direct electrophysiological monitoring of pharmacological SOCE inhibition and distinguishing between direct and indirect SOCE inhibition [25], and has prompted the widespread use of fluorescence based Ca^2+^ imaging. Ca^2+^ microscopy, on the other hand, whilst providing the level of sensitivity required to measure low levels of SOCE, is, however, incompatible with simultaneous controlling of the membrane potential. This, in turn, may explain the variety of effects observed upon treatment of GBM with SOCE inhibitors and cell proliferation [31,32]. Our data—showing almost complete inhibition of SOCE in the GBM cell lines A172, LN-18 and U-87 MG by 1 and 10 µM Synta66 in combination with the results obtained with Orai1 pore mutants—are consistent with direct inhibition of Orai1 by Synta66.

Synta66-mediated inhibition of SOCE in A172, LN-18 and U-87 MG was not, however, found to significantly affect cell viability. Growth of LN-18 and U-87 MG cells was, instead, perceptibly, though insignificantly, enhanced by 10 µM Synta66, whilst no significant effect of the compound was determined in migration assays. These findings, nonetheless, are at variance with previous reports of a critical role for SOCE-mediated GBM cell proliferation based on the treatment of U-251 MG and C6 cells with SKF-96365, 2APB and DES [31] and 2APB in D54, U-251 MG, U373, U-87 MG [32]. The latter inhibitors could disrupt store-independent Ca^2+^ pathways or other signalling mechanisms required for GBM cell proliferation in addition to a disturbance of SOCE.

## 4. Materials and Methods 

### 4.1. Cell Culture

U-87 MG cells (Medical University of Graz cell culture core facility) were cultivated in Eagle’s Minimum Essential Medium (EMEM) supplemented with 2 mM L-glutamine, 1 mM sodium pyruvate, 0.1 mM non-essential amino acids (NEAA) and 10% foetal bovine serum (FBS). A172 cells (Medical University of Graz cell culture core facility) were cultivated in Dulbecco’s Modified Eagle’s Medium (DMEM) supplemented with 10% FBS. LN-18 cells (ATCC, Manassas, VA, USA) were cultivated in DMEM supplemented with 5% FBS. All cell lines were cultured for a maximum of ten passages following the recovery of frozen cells. All media and supplements were obtained from Thermo Fisher Scientific, Waltham, MA, USA.

### 4.2. Compounds

Temozolomide (Merck Millipore, Burlington, MA, USA) was stored as powder at 4 °C and dissolved in the corresponding culture media immediately before use. Synta66 (Merck Millipore, Burlington, MA, USA) was stored as a 50 mM stock solution in DMSO at -80°C until dilution to a maximum DMSO concentration of 0.1% in cell culture experiments.

### 4.3. Electrophysiology

HEK293 cells were co-transfected (Transfectin, Biorad, Hercules, CA, USA) with the CFP-STIM1 and YFP-Orai1 constructs (1 μg + 1 μg). Electrophysiological measurements were performed 24 h after transfection, using the patch-clamp technique with a whole-cell recording configuration at 20 °C–24 °C using an Ag/AgCl reference electrode. Comparative electrophysiological measurements of Orai1 wild-type and mutants were carried out in paired comparisons on the same day. For STIM1/Orai1 current measurements, voltage ramps were applied every 5 s from a holding potential of 0 mV, covering a range of −90 to +90 mV over 1 s. The internal pipette solution for passive store-depletion consisted of (in mM) 145 caesium methane sulfonate, 8 NaCl, 3.5 MgCl_2_, 10 HEPES, 20 EGTA, pH 7.2. The extracellular solution contained (in mM) 145 NaCl, 5 CsCl, 1 MgCl_2_, 10 HEPES, 10 glucose, 10 CaCl_2_, pH 7.4. Na^+^-DVF solution comprised (in mM) 150 NaCl, 10 HEPES, 10 glucose, and 10 EDTA, pH 7.4. Applied voltages were corrected for the liquid junction potential, which was determined as +12 mV. Calcium release-activated channel (CRAC) currents shown were leak-corrected, either by subtraction of the initial trace or of the trace following inhibition by 10 μM La^3+^ at the end of the experiment. The expression patterns and levels of each Orai1 construct were carefully monitored by confocal fluorescence microscopy and found not to differ significantly.

### 4.4. Confocal FRET Microscopy 

A CSU-X1 Real-Time Confocal System (Yokogawa Electric Corporation, Vienna, Austria) connected with two CoolSNAP HQ2 CCD cameras (Photometrics) and a dual port adapter (dichroic: 505lp, cyan emission filter: 470/24, yellow emission filter: 535/30, Chroma Technology Corporation, Brattleboro, VT, USA) was used to record fluorescence images. This system was attached to an Axio Observer Z1 inverted microscope (Zeiss, Oberkochen, Germany) with two diode lasers (445 and 515 nm, Visitron Systems, Puchheim, Germany) and placed on a Vision IsoStation anti-vibration table (Newport Corporation, Irvine, CA, USA). The VisiView software package (V.2.1.4, Visitron Systems, Puchheim, Germany) was used for confocal system control and image generation. Illumination times for CFP/FRET and YFP images that were recorded consecutively with a minimum delay of about 300 ms. Image correction to address cross-talk and cross-excitation was performed before the calculation. To this end, appropriate cross-talk calibration factors were determined for each construct on each day of the FRET experiment. After threshold determination and background subtraction, the corrected FRET (Eapp) was calculated on a pixel-to-pixel basis with a custom-made software integrated into MATLAB (v7.11.0, The MathWorks, Inc., Natick, MA, USA) according to the method published by Zal and Gascoigne, with a microscope-specific constant G value of 2.75. All experiments were performed at room temperature. For the evaluation of FRET values, protocols performed in previous research studies [11,12] were used. YFP-STIM1 + CFP STIM1 as well as Orai1-YFP + STIM1-CFP transfected HEK-293 cells were grown on coverslips for 24 h and subsequently transferred to an extracellular solution consisting of 140 mM NaCl, 5 mM KCl, 1 mM MgCl_2_, 2 mM CaCl_2_, 10 mM glucose, and 10 mM HEPES buffer (adjusted to pH 7.4 with NaOH) +/− 10µM Synta66. SOCE was triggered using 1 µM TG in 0 mM Ca^2+^ solution +/− 10µM Synta66.

### 4.5. Calcium Imaging

Cells were cultivated on glass cover slips to 80 % confluency and then loaded with 1 µM FURA-2-AM (Sigma-Aldrich, St. Louis, MO, USA) in Calcium-free Tyrode buffer (0.1 mM EDTA, 138 mM NaCl, 1 mM MgCl_2_, 5 mM KCl, 10 mM HEPES, 10 mM Glucose, pH 7.4) for 40 min at room temperature (RT) in the dark, after which they were washed and incubated in Calcium-free Tyrode buffer ± 10 µM Synta66 for 10 min at RT in the dark. During the recordings (Olympus IX71) using Live Acquisition v2.6 software (FEI, Planegg, Germany), cells were excited alternately using 340/26 and 380/11 nm filters (Semrock, Rochester, NY, USA) in an Oligochrome excitation system (FEI), and fluorescent images were captured using 510/84-nm emission filter (Semrock, Rochester, NY, USA) with an ORCA-03G digital CCD camera (Hamamatsu, Herrsching am Ammersee, Germany). The 340 nm/380 nm ratio was used as an index of cytosolic Ca^2+^ levels. In time-course experiments, fluorescent cells were recorded initially in a Calcium-free Tyrode solution followed by addition of 30 µM BHQ. Subsequently, fluorescent cells were recorded in a 2 mM Calcium Tyrode solution with BHQ. Analogous experiments were recorded in the presence of 10 µM Synta66.

### 4.6. Western Blotting

Cells grown to 80% confluency in 10 cm dishes were harvested in phosphate-buffered saline (PBS) on ice with a cell scraper, pelleted by centrifugation (500 rpm, 4 °C, 6 min) and washed 2x with PBS before lysis in 700 µL of ice-cold lysis buffer (0.5% Nonidet P-40, 20 mM Tris-HCl, 100 mM NaCl, 2 mM EDTA, 10% glycerol) for 30 min on ice with pipetting. Lysed cells were then centrifuged for 10 min, 4 °C, 14,000 rpm and supernatant protein contents estimated by BCA Assay (Thermo Fisher Scientific, Waltham, MA, USA). For Orai1 Western blotting, lysates (20 µg) were de-glycosylated using a PNGase F kit (New England Biolabs, Ipswich, MA, USA), according to the manufacturer’s protocol. For STIM1 Western blots, standard lysates were used. (De)glycosylated lysates were, afterwards, separated by SDS-PAGE, and transferred to a nitrocellulose membrane, which was then blocked with 5% ovine serum albumin (BSA) in Tris-buffered saline with Tween20, (TBST) for 1 h. Primary antibodies were diluted in blocking buffer (Orai1 from Sigma-Aldrich, St. Louis, MO, USA at 1:2000 dilution, vinculin from Sigma-Aldrich, St. Louis, MO, USA at 1:5,000 dilution, STIM1 from Abcam, Cambridge, UK at 1:1000 dilution, glyceraldehyde 3-phosphate dehydrogenase (GAPDH) from Sigma-Aldrich, St. Louis, MO, USA at 1:1000 dilution) and incubated with the membranes o/n at 4°C. Visualization was performed by incubating with horseradish peroxidase-conjugated secondary antibodies and enhanced chemilumiscence (ECL) reaction (GE Healthcare Amersham, Chicago, IL, USA) on a ChemiDoc MP Imaging System (BioRad, Hercules, CA, USA). Band intensities were analysed using ImageJ.

### 4.7. Cell Viability (MTS Assay)

GBM cells were seeded in 96-well plates at a density of 4000 cells/well. After attachment, cells were treated with 1 µM or 10 µM Synta66 (0.1% DMSO final concentration) or 0.1% DMSO alone. Cell viability was assessed after 24, 48 and 72 h using an MTS assay (CellTiter 96 AQueous One Solution Cell Proliferation Assay, Promega, Walldorf, Germany) according to the manufacturer’s protocol. The converted chromogen was detected at 490 nm using a plate reader (CLARIOstar, BMG Labtech, Ortenberg, Germany).

### 4.8. Cell Growth (Hoechst Assay)

GBM cells were seeded in flat-bottomed black-walled 96-well plates (Corning, Corning, NY, USA) at a density of 8000 cells/well in 200 µL media and, after cell attachment, treated with 1 or 10 µM Synta66 (0.1% DMSO) or 0.1% DMSO alone for 0, 24, 48 and 72 h. Nuclei were quantitated by Hoechst staining (incubation in 6 µg/mL Hoechst 33258 in PBS for 15 min at RT in the dark followed by two PBS wash steps) and plate reading (CLARIOstar, BMG Labtech, Ortenberg, Germany) using 350 nm excitation and 455 nm emission wavelengths.

### 4.9. IC_50_ with Synta66 and TMZ

IC50 curves were generated from U-87 MG and A172 cells seeded at a density of 4000 cells/well and LN-18 cells at a density of 2000 cells/well in 96-well plates. Adherent cells were treated for 72 h with freshly diluted TMZ (0, 50, 100, 500, 1000, 2000, 5000, 7000, 10,000, 12,500, 15,000 and 20,000 µM) and cell viability determined after 24, 48 and 72 h using a CellTiter 96® AQueous One Solution Cell Proliferation Assay (MTS) (Promega, Walldorf, Germany) according to the manufacturer’s protocol. Plates were read on a CLARIOstar plate reader at 490 nm. Sigmoidal curve fitting was done with GraphPad Prism 8 using a variable slope with four parameters and the following equation: Y = Bottom + (Top-Bottom)/(1+(IC50/X)^HillSlope).

### 4.10. Scratch Assay

LN-18 cells were seeded into 12-well plates and fed with Dulbecco’s modified Eagle Medium supplemented with 4.5 g/l glucose and 2 mM L-glutamine (Gibco^TM^, Thermo Fisher Scientific, Waltham, MA, USA), 0.1 mM non-essential amino acids (Lonza, Basel, Swiss), 1% Penicillin/Streptomycin (Gibco^TM^, Thermo Fisher Scientific, Waltham, MA, USA) and 1% FBS (Sigma-Aldrich, St. Louis, MO, USA) (control) or medium supplemented with 10 µM Synta66 (Merck Millipore, Burlington, MA, USA) and grown to confluence. Scratches were created at the centre of confluent cell monolayers using a 1000 μL pipette tip. Floating cells were afterwards removed, and the resultant cell gap monitored with a Cell IQ V2 MLF Cell Imaging and Analysis System (Imagen, MA, USA) for 70 h (t1-t70).

### 4.11. Synta66 Docking and MD Simulations

The substrate (Synta66) is minimized using LigPrep tool in Schrodinger [65]. The protein Preparation wizard is used for receptor grid generation using OPLS-2005 [66] force fields. The receptor grid generation and docking are performed using the GLIDE tool [67] of Schrodinger. The possible active binding site for Synta66 is given during receptor grid generation between two adjacent subunits of Orai channel. 

The conformation corresponding to the highest scoring was then used as a starting point for MD simulations. Two different protocols were followed. In the first case, systems were prepared with either one or three molecules of Synta66 already present in the pocket identified by docking. In the second case, three molecules of Synta66 were present at the entry of the pore yet not in contact with the protein (10 Å away from the selectivity filter). These simulations were intended to check if molecules of Synta66 would naturally interact with the identified pocket. A potential barrier was applied to simulations with free ligands to prevent any of the molecules to diffuse toward the bulk water. This potential was restraining the molecules to diffuse from a distance farther than 15 Å from the centre of mass of the selectivity centre. The potential was applied by making use of the PLUMED2 [68] software. The CHARMM36 forcefield was used for the simulations, protein [69] and lipids [70]. The water model used is TIP3P [71]. The ionic strength used in all the simulations corresponds to a concentration of NaCl at 0.15 M. For the ions, the scaled parameters were used for both sodium and chloride [72]. The systems were prepared by using the CHARMM-GUI web server [73] and the membrane builder input generator [74]. The box size is 132.35 Å × 132.35 Å × 120.89 Å. The Charmm CgenFF [75] force field was used for the Synta66 molecule and parameters obtained through the use of the CgenFF interface (https://cgenff.umaryland.edu) [76].

All simulations were performed using GROMACS 5.1.4 [77]. The temperature and pressure were maintained at 310 K and 1.013 atm using the Nove–Hoover [78,79] and Parrinello–Rahman [80,81] algorithms. Short-range interactions of Lennard–Jones were cut off at 1.2 nm. Long-range electrostatic interactions were treated using particle-mesh Ewald [82] with a real-space cut-off at 1.2 nm. Covalent hydrogen bonds were constrained using an LINCS algorithm [83,84]. The integration time-steps was 2 fs.

## 5. Conclusions

In conclusion, the present study showed Synta66 to be (i) a selective pore inhibitor of the Orai1 channel that mediates its effects by binding close to the selectivity filter, and (ii) an efficient inhibitor of SOCE in GBM cells. Synta66 action on the Orai1 pore was determined by complete inhibition of ectopically expressed STIM1/Orai1 currents, whilst Orai1 mutants exhibiting reduced Ca^2+^ selectivity were less sensitive to Synta66 inhibition. No significant inhibition of GBM cell viability by Synta66 was observed. Synta66 is thus a valuable tool for the evaluation of the contribution of Orai1 channels to cancer cell progression. 

## Figures and Tables

**Figure 1 cancers-12-02876-f001:**
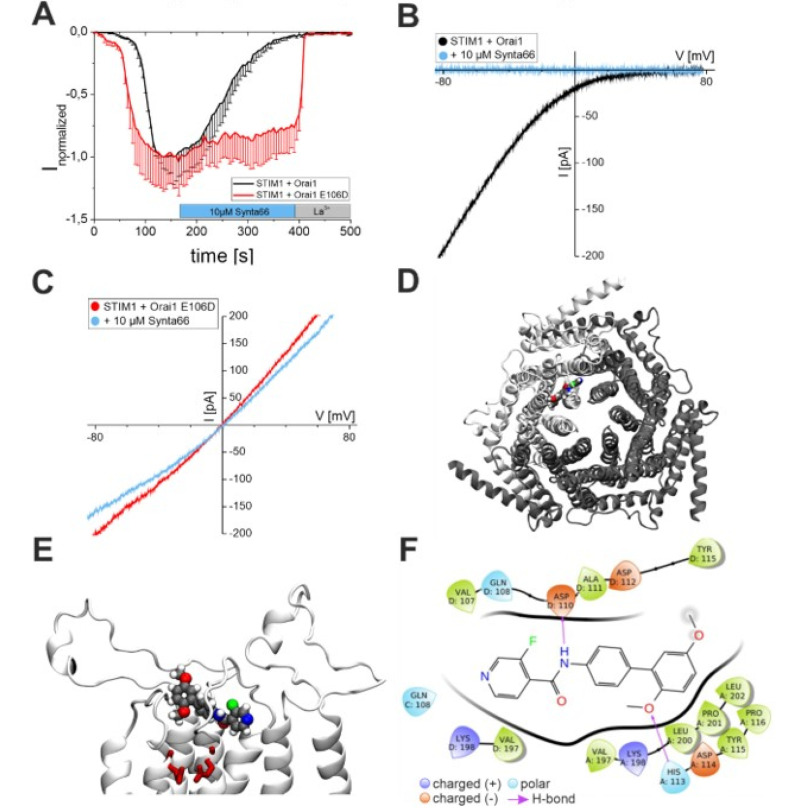
The inhibitory action of Synta66 is affected by Orai1 pore geometry: (**A**) Time course of normalized whole-cell inward rectifying currents at −86 mV, maximally activated upon passive store depletion of HEK293 cells co-expressing STIM1 and Orai1 (black) or Orai1 E106D (red), upon perfusion of 10 μM Synta66 and subsequent block by 10 μM La^3+^ (n = 6 − 10 cells, from at least two individual transfections). (**B**,**C**) Corresponding I/V relationships of STIM1 and Orai1 (black, **B**) or Orai1 E106D (red, **C**) after maximal store-operated activation and upon addition of 10 μM Synta66 (blue). (**D**,**E**) Top view (**D**) and side-view (**E**) of the Orai channel showing docking configuration of Synta66 with the most negative glide score. In (**E**), Synta66 is shown as spheres, the selectivity filter is highlighted as red sticks. (**F**) Synta66 in docking position 1. Black lines resemble the protein backbone. Two H-bonds are formed between Synta66 and amino acids of Orai at Asp110 and His113.

**Figure 2 cancers-12-02876-f002:**
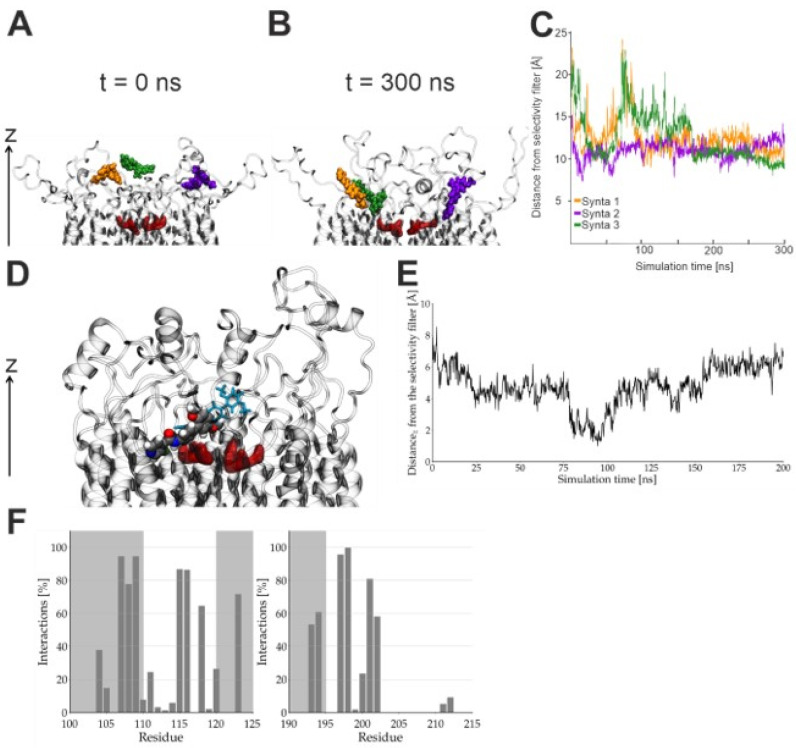
Interactions between free ligands with human Orai1: (**A**,**B**) Starting configuration (**A**, t = 0 ns) and (**B**) MD simulations after 300 ns with three molecules of Synta66 present at the entry of the pore. The Orai channel protein is represented as a grey glassy ribbon with the selectivity filter highlighted in red. Individual molecules of Synta66 are represented in orange, violet and green. (**C**) Projection of the distance along the z-axis of the distances between the centre-of-mass of each molecule of Synta66 (colour code as in **A**,**B**) and the selectivity filter as a function of the simulation time. (**D–E**) Molecular Dynamic (MD) simulations of a single docked Synta66 compound with start position of pose 1 (stick model) and after 200 ns of simulations (sphere model). (**e**) Projection of distances of a single Synta66 to the selectivity filter as in (**C**). (**F**) Interactions (in %) for a single Synta66 with Orai1 residues.

**Figure 3 cancers-12-02876-f003:**
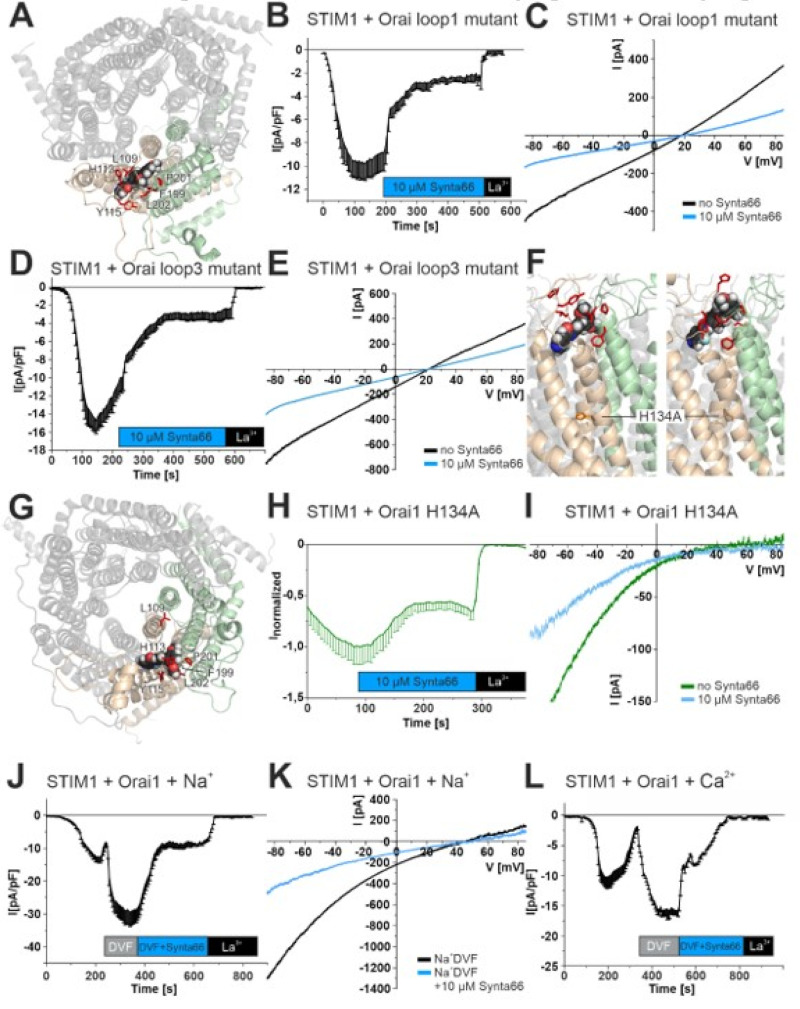
Synta66 inhibition is impaired in Orai1 mutants that yield non-selective permeation: (**A**) Snapshot of Orai1 channel (top view) with a single Synta66 bound to loop1 and loop3 residues of Orai1. Residues that are mutated in (**B–E**) are highlighted as red sticks. (**B**,**D**,**H**) Time course of store-operated whole-cell currents at −86 mV for a co-expression of STIM1 and (**B**) Orai1 loop1 mutant (L109D, H113G, Y115G), (**D**) Orai1 loop3 mutant (F199G, P201G, L202G) or (**H**) Orai H134A mutant. Upon maximum Orai1 activation, 10 µM Synta66 was added to the extracellular solution followed by 10 µM La^3+^. (**C**,**E**,**I**) Current voltage relationships from representative experiments from (**B**,**D**,**H**) upon maximal store-operated Ca^2+^ activation and of steady state currents upon perfusion with a 10 µM Synta66 containing solution. (**F**) Side view of the wild-type Orai1 channel (left) and Orai1-H134A mutant (right) with residues from (**A**) and H134/A134 highlighted as stick model. (**G**) Structure of the Orai1-H134A simulations (top view) with residues from (**A**) highlighted. (**J**,**L**) Time course of store-operated whole-cell currents at −86 mV by STIM1 and Orai1 in a standard Ca^2+^ based extracellular solution. After reaching a maximum current plateau, extracellular solution was substituted to a Na^+^ based divalent ion free solution (Na^+^DVF). Currents were subsequently monitored in a (**J**) Na^+^DVF with 10 µM Synta66 or (**L**) Ca^2+^ based extracellular solution with 10 µM Synta66. (**K**) Current voltage relationships from representative experiments from (I) in a Na-DVF solution (black) or with 10 µM Synta66 (blue).

**Figure 4 cancers-12-02876-f004:**
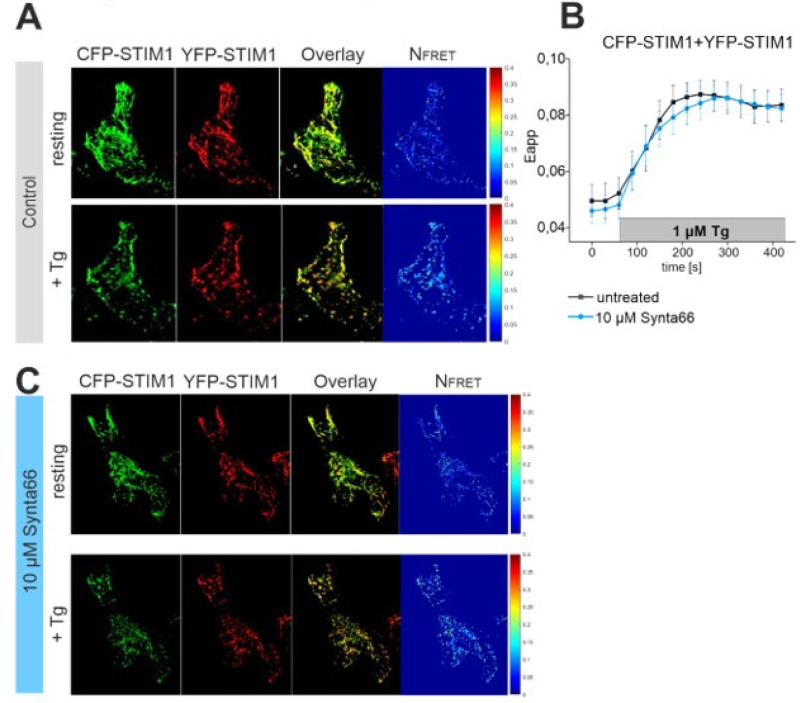
STIM1 interaction is not affected by Synta66**:** (**A**) Localization, overlay and calculated fluorescence resonance energy transfer (FRET) life cell image series of cyan fluorescent protein (CFP)-STIM1 and yellow fluorescent protein (YFP)-STIM1 in the absence (upper panel) and presence of Thapsigargin (TG, 1 µM) mediated endoplasmic reticulum (ER) Ca^2+^ store-depletion (lower panel). (**B**) Time-course of calculated FRET between CFP- and YFP-STIM1 with and without pre-treatment with 10 µM Sytna66. ER Ca^2+^ store-depletion was induced by 1 µM TG. (**C**) Analogous experiments as in (**A**) upon incubation with 10 µM Synta66 for 20 min.

**Figure 5 cancers-12-02876-f005:**
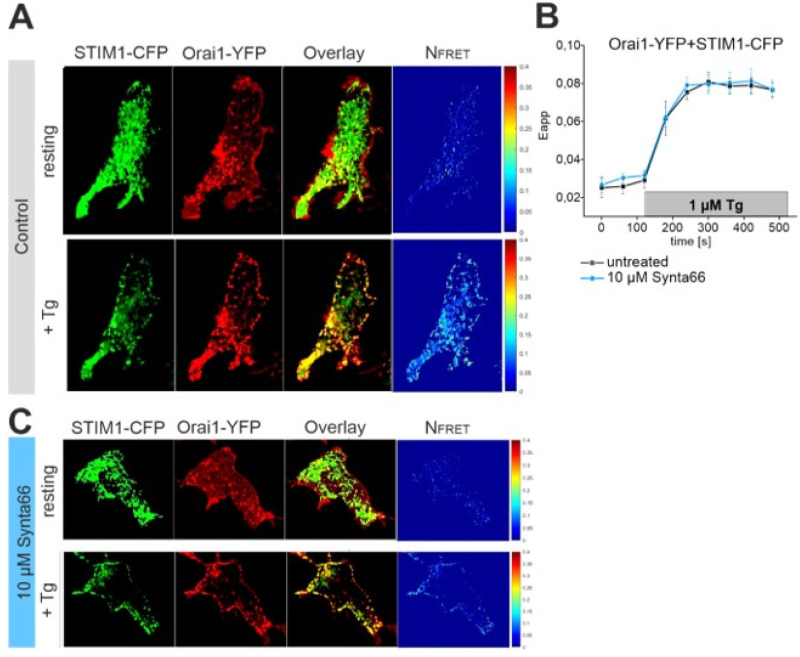
STIM1-Orai1 interaction is not affected by Synta66: (**A**) Localization, overlay and calculated FRET life cell image series of STIM1-CFP and Orai1-YFP in the absence (upper panel) and presence of 1 µM TG mediated ER Ca^2+^ store-depletion (lower panel). (**B**) Time-course of calculated FRET between STIM1-CFP and Orai1-YFP with and without treatment with 10 µM Sytna66. ER Ca^2+^ store-depletion was induced by 1 µM TG. (**C**) Analogous experiments as in (**A**) upon pre-treatment with 10 µM Synta66 for 20 min.

**Figure 6 cancers-12-02876-f006:**
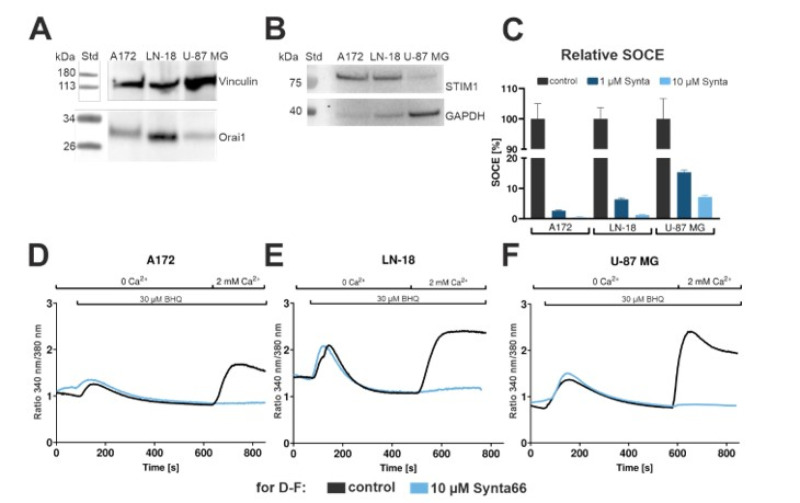
Inhibition of SOCE in GBM cell lines by Synta66: (**A**) Western blot for detection of endogenously expressed Orai1 in lysates of A172, LN-18 and U-87 MG cells. Lysates were deglycosylated in order to achieve a single Orai1 band as in [51]. (**B**) Western Blot for detection of endogenously expressed STIM1 in lysates of A172, LN-18 and U-87 MG cells. (**C**) Normalized maximal values of SOC activation from Fura-2 AM loaded GBM cells (A172, LN-18 and U-87) and comparison to SOC treated with 1 and 10 µM Synta66 (light and dark blue, respectively). Results shown as mean ± SEM from two independent experiments (n = 109–192 cells). A172 and LN-18 had no detectable SOCE peak, U-87 MG had 7.0± 0.6 % SOCE peak. (**D–F**) Time course experiments show mean values of cytosolic Ca^2+^ measurements in Fura-2 AM loaded GBM cell lines. Cells are monitored initially in a Ca^2+^ free extracellular solution followed by application of 30 µM BHQ and addition of 2 mM Ca^2+^ to monitor SOCE (black). Analogous experiments with pre-treatment of 10 µM Synta66 immediately before the start of the experiment were used (blue).

**Figure 7 cancers-12-02876-f007:**
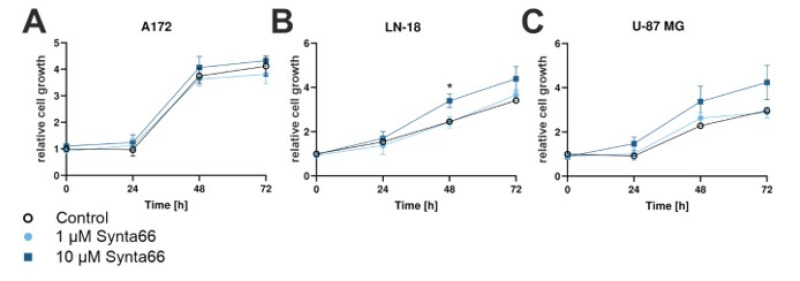
Effect of Synta66 on cell growth: (**A**–**C**) Effect of Synta66 (1 and 10 µM, light and dark blue respectively) on proliferation of GBM cell lines (**A**) LN-18, (**B**) A172 and (**C**) U-87 MG was examined via Hoechst staining of nuclei. Signal was normalized to control signal at t = 0 h, results shown as mean ± SEM, n = 6 from two independent experiments; *: *p*-value < 0.05 (two-sided *t*-test).

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
