# Peer review of "Blockage of Store-Operated Ca^2+^ Influx by Synta66 is Mediated by Direct Inhibition of the Ca^2+^ Selective Orai1 Pore"

_cancers, 2020, doi:10.3390/cancers12102876_

Round 1
Reviewer 1 Report
The manuscript entitled “Blockage of store-operated Ca2+ influx by Synta66 is mediated by direct inhibition of the Ca2+ selective Orai1 pore” attempts to evaluate the mechanism of action for Synta66. Although the effectiveness has been shown already in multiple studies the mechanism of action has not yet been fully clarified. This paper by Waldherr et al. follows up on an earlier report (partially) from this group where the action of selective CRAC channels blockers has been investigated (Derler et al 2013, Cell calcium). In the previous paper, they already concluded that the action of blockers is affected by the Orai pore geometry, at least for the GSK compounds. Additionally, they reported that Synta66 can block, beside the Orai1 mediated currents, also Orai3 mediated currents even with a faster t1/2 of 50 s instead of 75 s for Orai1 (Derler et al 2013, Cell calcium).
However, to my surprise, the manuscript completely omitted to discuss the fact rather than conduct experimental procedures concerning the Orai3 subunits. The pore structure of Orai1 and Orai3 channels, and especially the TM1 and TM3 helices, are quite similar, and therefore at least a discussion is inevitable. However, the Loop1 and Loop3 region between Orai1 and Orai3 show clear differences. Besides, in cells (even glioblastoma) the composition of the channels does not consist of only Orai1 channels itself, but rather we are facing a situation where multiple subunits contribute to pore formation. This situation has not been mention at all rather than discussed in the entire manuscript. Overall, while these findings are interesting, lack of proper discussion and experiments limits enthusiasm.
I have several major concerns to rise:
1) The discussion section needs a major revision (please see details above). Additionally, some of the findings in this manuscript contradict previous reports a detailed explanation would be desirable at this point. What are the reasons for the discrepancies? One of the reasons could be the cell lines itself used in the studies (high or low-grade glioblastoma) rather than a disruption of other signaling mechanisms.
2) Section 2.5 about the similarities of CM-4620 to Syntax66 is completely based on stimulation. In my opinion, the authors need to show supportive data or remove the section completely. Most journals do not permit “unpublished results” comments. Please confirm that this is acceptable for Cancers. On the other hand, this section can be removed easily without losing important information.
3) Why the authors used such a high concentration of Synta66 (10µM) in their most crucial experiments (all patch-clamp recordings with wt Orai1 and mutants, Figures 1 and 3)? Is the block concentration dependent? If so, additional experiments are necessary. The authors showed in their MD stimulation (Fig. 2) that one single Synta66 molecule can anchor deeper within the transmembrane domain and also closer to the selectivity filter (Figure 2D, E). So one could speculate that maybe more molecules (higher concentration?) hinder each other to reach their position. A dose-responds curve or at least set of patch-clamp experiments performed with 1 µM Synta66 could help to clarify this point. Also, it has already been reported that even lower concentrations of Synta66 (25 nM – 5µM) (Li J., et al, 2011, Br J Pharmacol) are sufficient for inhibition in other cell lines.
Furthermore, the same concentration is able to inhibit SOCE in cells but with differential potencies and that would speak against Orai1 as the target protein alone. This should be explained more carefully.
4) For the calcium recordings, the authors used high BHQ concentration, as a not very specific SERCA inhibitor with many sites of action beside SERCA. What is the reason not to use thapsigargin as the common SERCA blocker?
5) What is the role of the STIM proteins in this context? Is Synta66 able to affect STIM oligomerization or S1-O1 coupling? Suitable FRET experiments could provide the answer.
6) The western blots are not convincing (Figure 4A and B), especially for STIM1 protein, as the GAPDH is barely seen on the blot. For a better comparison, a densitometry analysis would be more suitable for both western blots.
Minor points:
1) The labeling of the amino acids in Figure 1F is hard to see.
2) Is there a reason why the currents have been extracted by -86 mV?
3) Please check the manuscript for typographical errors (for example line 87, …).
4) Figure 4D, E, F labeling of the BHQ is wrong 30 M instead of 30 µM
5) The experiments in Figure 6 with temozolomide don’t add any important information to clarify the Synta66 side of action and can be easily removed.
Author Response
The manuscript entitled “Blockage of store-operated Ca2+ influx by Synta66 is mediated by direct inhibition of the Ca2+ selective Orai1 pore” attempts to evaluate the mechanism of action for Synta66. Although the effectiveness has been shown already in multiple studies the mechanism of action has not yet been fully clarified. This paper by Waldherr et al. follows up on an earlier report (partially) from this group where the action of selective CRAC channels blockers has been investigated (Derler et al 2013, Cell calcium). In the previous paper, they already concluded that the action of blockers is affected by the Orai pore geometry, at least for the GSK compounds. Additionally, they reported that Synta66 can block, beside the Orai1 mediated currents, also Orai3 mediated currents even with a faster t1/2 of 50 s instead of 75 s for Orai1 (Derler et al 2013, Cell calcium). However, to my surprise, the manuscript completely omitted to discuss the fact rather than conduct experimental procedures concerning the Orai3 subunits. The pore structure of Orai1 and Orai3 channels, and especially the TM1 and TM3 helices, are quite similar, and therefore at least a discussion is inevitable. However, the Loop1 and Loop3 region between Orai1 and Orai3 show clear differences. Besides, in cells (even glioblastoma) the composition of the channels does not consist of only Orai1 channels itself, but rather we are facing a situation where multiple subunits contribute to pore formation. This situation has not been mention at all rather than discussed in the entire manuscript. Overall, while these findings are interesting, lack of proper discussion and experiments limits enthusiasm. I have several major concerns to rise:
1) The discussion section needs a major revision (please see details above). Additionally, some of the findings in this manuscript contradict previous reports a detailed explanation would be desirable at this point. What are the reasons for the discrepancies? One of the reasons could be the cell lines itself used in the studies (high or low-grade glioblastoma) rather than a disruption of other signaling mechanisms.
We thank the reviewer for addressing these important points. We have now discussed the structural details of Orai1, Orai2 and Orai3 and also iso-form dependent inhibition of various inhibitors. We have now included in the discussion “ It is of note that inhibition of Orai channels by several SOC inhibitors, including Synta66, BTP2, GSK-7975A is dependent on the Orai isoform. A recent work determined that 10µM Synta66 inhibited Orai1, while Orai2 currents were even further stimulated and Orai3 currents were largely unaffected. We compared the amino-acid sequence within the three Orai isoform that were observed to dock with Synta66. L109, H113 and Y115 residues in Orai1 loop1, are conserved in Orai3, while H113 is a tyrosine in Orai2. F199, P201 in TM3 helix of Orai1 are fully conserved in Orai2 and Orai3. L202 in Orai1 is also a hydrophobic amino acid, with a valine in Orai2 and an isoleucine in Orai3. These sequence analysis shows that key residues are largely conserved, however, additional interaction sites with Synta66 with Orai2 or Orai3 residues may further tune Orai channel modulation.” (page 12 of 21, line 343-352).
We also stated now in the discussion “Concatenated Orai dimers further diminished Synta66 block (62) suggesting that the here investigated store-operated Ca2+ channels in GBM cell lines are largely based on Orai1 channels. A different heteromeric Orai composition may tune the correlation of SOCE inhibition by Synta66 in other GBM cell types. Moreover, a molecular switch from Orai1 homomers into Orai1-Orai3 heteromers resulted in store-independent currents and promoted cell proliferation.” (page 12 of 21, line 353-357).
We agree that “contradict” does not fit here, as Synta66 was not used before to interfere with GBM Ca2+ signaling and cell viability. Rather, the link between potential non-selective SOCE inhibitors and GBM cell progression has been addressed now more carefully. In the end of the discussion, we have now written more specifically and with more caution that “These findings, nonetheless, are at variance with previous reports of a critical role for SOCE-mediated GBM cell proliferation based on the treatment of U-251 MG and C6 cells with SKF-96365, 2APB and DES and 2APB in D54, U-251 MG, U373, U-87 MG. The latter inhibitors could reflect a disruption of other signalling mechanisms required for GBM cell proliferation in addition to a disturbance of SOCE.” (page 12 of 21, line 383-387).
2) Section 2.5 about the similarities of CM-4620 to Syntax66 is completely based on stimulation. In my opinion, the authors need to show supportive data or remove the section completely. Most journals do not permit “unpublished results” comments. Please confirm that this is acceptable for Cancers. On the other hand, this section can be removed easily without losing important information.
We agree with the reviewer and have now deleted section 2.5.
3) Why the authors used such a high concentration of Synta66 (10µM) in their most crucial experiments (all patch-clamp recordings with wt Orai1 and mutants, Figures 1 and 3)? Is the block concentration dependent? If so, additional experiments are necessary. The authors showed in their MD stimulation (Fig. 2) that one single Synta66 molecule can anchor deeper within the transmembrane domain and also closer to the selectivity filter (Figure 2D, E). So one could speculate that maybe more molecules (higher concentration?) hinder each other to reach their position. A dose-responds curve or at least set of patch-clamp experiments
performed with 1 µM Synta66 could help to clarify this point. Also, it has already been reported that even lower concentrations of Synta66 (25 nM – 5µM) (Li J., et al, 2011, Br J Pharmacol) are sufficient for inhibition in other cell lines. Furthermore, the same concentration is able to inhibit SOCE in cells but with differential potencies and that would speak against Orai1 as the target protein alone. This should be explained more carefully.
The work by Li J. et al. compared Synta66 inhibition in vascular smooth muscle cells and determined an IC50 of 25 nM while experiments in immune cells determined an IC50 1.7 µM. We have previously shown that that over-expressed STIM1/Orai1 currents were inhibited with an IC50 of ∼4 μM (Derler et al. Cell Calcium 2013). A similar inhibitory profile has been reported for Synta66 on RBL mast cells exhibiting an IC50 of ∼1-3 μM. We have now addressed these points in the results section 1.1. Additionally, we have performed patch clamp experiments of STIM1 and Orai1 expressing cells and show a slow inhibition by 1 µM Synta66 in the new Figure S1.
4) For the calcium recordings, the authors used high BHQ concentration, as a not very specific SERCA inhibitor with many sites of action beside SERCA. What is the reason not to use thapsigargin as the common SERCA blocker?
Indeed, we have used BHQ (e.g. Muik et al. EMBO J. 2011) and TG (e.g. Frischauf et al. Science Signaling 2016) previously to demonstrate that STIM1 and Orai1 mediated storeoperated Ca2+ entry could be stimulated by both drugs. To double check that also stimulation by 2 µM TG results in similar inhibition by 1 µM Synta66 we have redone experiments in U87 MG cells and determined identical results as with BHQ treatment.
5) What is the role of the STIM proteins in this context? Is Synta66 able to affect STIM oligomerization or S1-O1 coupling? Suitable FRET experiments could provide the answer.
We have now performed STIM1 – STIM1 FRET experiments in dependence of storedepletion with and without Synta66 pre-treatment. No significant difference was observed, and we now show this important set of experiments as new Figure 4. Also, we monitored STIM1 and Orai1 interaction in FRET experiments in dependence of store-depletion with and without Synta66 pre-treatment. No significant difference was observed, and we now show these experiments as new Figure 5.
6) The western blots are not convincing (Figure 4A and B), especially for STIM1 protein, as the GAPDH is barely seen on the blot. For a better comparison, a densitometry analysis would be more suitable for both western blots.
We have now performed new western blot experiments for STIM1 and GAPDH controls in Figure 6B and provide densiometric results in Figure S5.
Minor points: 1) The labeling of the amino acids in Figure 1F is hard to see.
We have now increased the size of the individual panels in Figure 1.
2) Is there a reason why the currents have been extracted by -86 mV?
We determine currents at -86mV to avoid capacitive currents at the beginning of the voltage ramp.
3) Please check the manuscript for typographical errors (for example line 87, …).
We have corrected line U-87 MG and double checked the manuscript.
4) Figure 4D, E, F labeling of the BHQ is wrong 30 M instead of 30 µM
We have now corrected the labels in Figure 4D-F.
5) The experiments in Figure 6 with temozolomide don’t add any important information to clarify the Synta66 side of action and can be easily removed.
We have now transferred the TMZ figure into the supplements (Figure S7).
Reviewer 2 Report
The studies performed by the authors give evidence for the mechanism of Synta66 inhibition of CRAC channels through MD simulation, electrophysiology and confocal microscopy through SOCe entry experiments. The authors have identified the residues through MD simulation that interact Synta66 interacts with in order to inhibit the calcium selective ORAI pore forming units.
Figure 1 takes from the authors paper from Derler et al., 2013, Cell Calcium ref 41 which looks at the same aspects that are being discussed in figure 1 in the current paper. Please clarify this.
Author Response
The studies performed by the authors give evidence for the mechanism of Synta66 inhibition of CRAC channels through MD simulation, electrophysiology and confocal microscopy through SOCe entry experiments. The authors have identified the residues through MD simulation that interact Synta66 interacts with in order to inhibit the calcium selective ORAI pore forming units.
Figure 1 takes from the authors paper from Derler et al., 2013, Cell Calcium ref 41 which looks at the same aspects that are being discussed in figure 1 in the current paper. Please clarify this.
We have redone the key experiment STIM1+Orai1 with Synta66 (Fig. 1A, black trace) similar as in Derler et al. 2013. This was important to compare and show it with the Orai1 pore mutants. We have now cited this paper in the sentence “Addition of 10 µM Synta66 completely blocked STIM1/Orai1 currents at the maximum activation point, (Figure 1A, black trace) in line with our previous results (Derler et al.).
Round 2
Reviewer 1 Report
The authors have adequately addressed my major and minor concerns.
However, I'd recommend checking the spelling before the final publication.
Author Response
We thank the reviewer. We have now double checked for spelling errors and the english language was corrected by an native speaker.